# Learning, Forgetting, Remembering:
# Insights From Tracking LLM Memorization During Training

**Danny D. Leybzon**
Universitat Pompeu Fabra
danny.leybzon@gmail.com

**Corentin Kervadec**
Universitat Pompeu Fabra
corentin.kervadec@gmail.com

## Abstract

Large language models memorize portions of their training data verbatim. Our findings indicate that models exhibit higher memorization rates both early on and at the very end of their training, with the lowest rates occurring midway through the process. This phenomenon can be attributed to the models retaining most of the examples memorized early on, while forgetting many more examples as training progresses. Interestingly, these forgotten examples are sometimes re-memorized later on, often undergoing cycles of forgetting and re-memorization. Notably, examples memorized early in training are more likely to remain consistently retained, suggesting that they become more firmly 'crystallized' in the model's representation. Based on these insights, we tentatively recommend placing data that is more likely to be sensitive in the middle stages of the training process.

## 1 Introduction

Large language models (LLMs) can achieve state-of-the-art results on a variety of NLP tasks (Liang et al., 2023) but are not without their problems. One such problem is their propensity to output portions of their training data verbatim, a phenomenon referred to as "memorization" (Carlini et al., 2019).

Memorization in LLMs is a potentially undesirable outcome because it can lead to the unintentional disclosure of private information such as personal data (including credit card or social security numbers), trade secrets, passwords, etc. (Carlini et al., 2019). Training data extraction attacks seek to extract training examples from a model verbatim and memorization enables these types of attacks to succeed (Carlini et al., 2021; Nasr et al., 2023). By better understanding why memorization occurs, researchers will be able to minimize the memorization of sensitive information and mitigate the risk of extraction attacks (Huang et al., 2022).

Previous work (Biderman et al., 2023) (discussed in Section 2) has concluded that LLMs

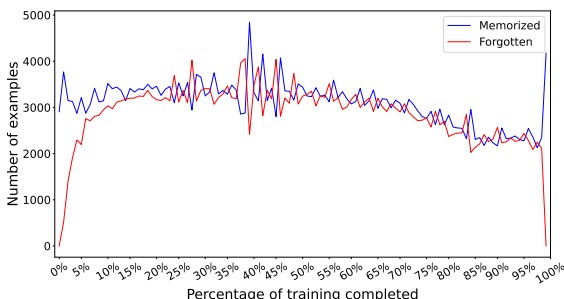

Figure 1: We decompose memorization into *newly memorized* and *forgotten* examples at each training checkpoint. The blue line represents the number of examples that are newly memorized compared to the previous checkpoint, while the red line indicates the number of previously memorized examples that are forgotten. The difference between these two lines reflects the overall change in memorization.

memorize a fixed proportion of their data at each step and, as a result, has avoided making recommendations about the order in which data is fed to the model throughout training. We find that:

1. Models tend to memorize a **higher** proportion of their training data **early** on during training

2. Discrepancy in memorization rate is caused by the number of examples **forgotten** by the model at each step, while the number of **newly memorized** examples stays **nearly constant**

3. But forgotten examples get **re-memorized** throughout training at a very high frequency

4. This re-memorization occurs even if examples have been **markedly forgotten**

5. Examples memorized **early on** in training are **more likely** to remain memorized throughout the **entire** training process

As a result, we tentatively recommend model developers to put the data that is most likely to be sensitive in the middle of the training process.

## 2 Background

**Defining Memorization in Language Modeling**
The standard definition of memorization used in this paper comes from Carlini et al. (2021), which introduces a quantifiable definition of "k-eidetic memorization":

A string $s$ is $k$-eidetic memorized (for $k \geq 1$) by an LM $f_\theta$ if $s$ is extractable from $f_\theta$ and $s$ appears in at most $k$ examples in the training data $X$:

$$|\{x \in X : s \subseteq x\}| \leq k. \tag{1}$$

Key to the definition of memorization is "extractability", which refers (Carlini et al., 2023) to the ability to prompt a model to generate a string given a text prompt of length k which precedes the target string in the training data. More concretely: A string $s$ is *extractable with $k$ tokens of context* from a model $f$ if there exists a (length-$k$) string $p$, such that the concatenation $[p \parallel s]$ is contained in the training data for $f$, and $f$ produces $s$ when prompted with $p$ using greedy decoding.[1]

All strings that are extractable in such a way are counted as memorized. Indeed, extractability acts as a highly sensitive "canary in the coal mine" for other, more harmful forms of memorization, like the ones taken advantage of in training data extraction attacks (Nasr et al., 2023). If training data is extractable via prompting the model with training data extracts, it is possible that other attack vectors will also allow sensitive training data extraction.

**Memorization Training Dynamics** Previous work on the training dynamics of memorization in language models has primarily been motivated by preventing memorization or getting early signals of it during training. Memorization rates have been found to scale with parameters such as model size (Carlini et al., 2023; Tirumala et al., 2024; Biderman et al., 2024), the frequency of appearance of the example in the dataset (Carlini et al., 2023; Hernandez et al., 2022), the length of the context k used to prompt the model (Carlini et al., 2023), and the learning rate (Tirumala et al., 2024).

Previous research on the impact of training order on memorization found that memorization is well-modeled by a Poisson distribution, indicating that memorization is approximately equally likely to happen at each step in the training process (Biderman et al., 2023). Further research found little cor-

---

[1] Note that the variable "k" is used differently in these two definitions.

relation between the examples memorized throughout the training process, indicating that the model is forgetting many of the examples it had previously memorized and then re-learning them seemingly at random (Biderman et al., 2024). These findings are in contradiction to the phenomena that we observe in our analysis.

**Forgetting in Language Modeling** Few studies have discussed "forgetting" in the context of LLM memorization research. Most memorization research we surveyed is not focused on the training dynamics of memorization and the ones focused on training dynamics (Biderman et al., 2023, 2024) did not discuss forgetting. A notable exception is (Tirumala et al., 2024), where the authors find a logarithmic forgetting curve that ultimately comes to a stable "forgetting baseline", primarily dictated by model size.

**OLMo** Our model of choice in this work is the 7 billion parameter Open Language Model (OLMo) (Groeneveld et al., 2024) published by the Allen Institute for Artificial Intelligence. OLMo is a framework that consists of trained OLMo models, the pre-training dataset Dolma (Soldaini et al., 2024), and various other artifacts. The OLMo models are decoder-only LLMs that have been trained using similar practices to the currently available, state-of-the-art LLMs and are competitive with those LLMs in many of the OLMo authors' evaluations. This makes them an ideal proxy for evaluating memorization and forgetting in those state-of-the-art LLMs, which we can not evaluate directly because they do not follow the same open framework as OLMo. We reproduce all of our experiments with the Pythia model suite, in Appendix A.

## 3 Methodology

To study the impact of training order on memorization, we extracted and then deduplicated 64-token sequences from OLMo's training dataset. We then passed the first 32 tokens of these sequences to evenly-spaced checkpoints throughout OLMo's training process and had these checkpoints generate 32 more tokens. We compared these generated tokens with the "ground truth" (i.e. the last 32 tokens in the original extractions) to evaluate whether and to what extent the sequence had been memorized.

**Sequence Extraction** The version of OLMo used in this paper was trained on version `v1_5-sample` of the Dolma dataset (Soldaini et al.,

2024). This corpus is split into 2,418 files, each of which contains a list of documents sorted by their source. For each file, we extracted the first 500 documents that had a length greater than or equal to 64 tokens and extracted the first 64 tokens from each document. We chose to extract the first 64 to reproduce the work in (Biderman et al., 2023), where the length 64 is chosen arbitrarily and the first tokens are extracted to minimize covariate effects. This resulted in a dataset of 1,208,000 sequences of length 64, where each sequence appeared at the beginning of a document in Dolma. Two files did not have any documents with lengths greater than 64, which explains the 1,000 sequence discrepancy between our final sequence count and the expected final sequence count of 1,209,000 ($2,418 * 500 = 1,209,000$).

**Deduplication** Prior research has shown that repeated examples in the training data are more likely to be memorized (Carlini et al., 2023; Hernandez et al., 2022; Lee et al., 2022; Kandpal et al., 2022). Although the Dolma dataset that OLMo was trained on has been heavily deduplicated, some sequences repeat in various places in the training data. To minimize the impacts of often-repeated sequences on our analysis, we deduplicated our dataset before performing our analysis.[2]

**Response Generation** After deduplicating our data, we split each sequence into two 32-token subsequences. We selected 112 checkpoints separated by 5,000 training steps each, starting from step0-tokens0B (which represents the randomly initialized model that has been exposed to no training data) to step555000-tokens2455B (which represents the fully-trained model that has been exposed to approximately 2,455,000,000 tokens). We passed the first 32-token subsequence prompts to the model and generated 32-token responses using greedy decoding, following the standard definition of extractability (Carlini et al., 2023). During generation, we used the default HuggingFace function parameters, except for using 16-bit quantized versions of the checkpoints and running the generations on our GPUs. We used batches of size 32.

**Memorization Evaluation** We evaluated whether a checkpoint had memorized a given sequence by directly comparing the 32-token sequence generated by the model against the

original 32-token response we had extracted from the training dataset. If the generated sequence exactly matched the ground truth sequence, we counted that sequence as a "memorized" example for that checkpoint.

# 4 Results

## 4.1 Descriptive Statistics

Of the 1,2080,000 sequences extracted from OLMo's training data in Sequence Extraction, 1,205,572 remained after deduplication. Of these, 44,559 were memorized by at least one of the 112 OLMo checkpoints we considered. Hence, **3.7% of the sequences have been memorized at least once during the training**. The step0, randomized model had memorized zero sequences, while the final model had memorized 26,423 (2.19% of all sequences)[3]. There were 1,127 (0.09% of the total) sequences memorized at every checkpoint we evaluated, excluding the step0 checkpoint.

The fact that only 0.09% of examples are memorized by every checkpoint demonstrates an important insight in this work: LLMs memorize their training data but then forget parts of it throughout the training process. As further analysis will demonstrate (Section 4.2 and 4.5) sometimes examples are memorized, forgotten, and then re-memorized again in subsequent checkpoints.

## 4.2 Memorization Trends at Completion

Model developers and researchers may be particularly interested in understanding the examples that the final checkpoint (i.e. the model at the end of the training process) has memorized. This might be of particular interest because this checkpoint represents the model that will either be deployed directly to users or fine-tuned and then deployed. With that in mind, we start our analysis by looking at only examples memorized by the final checkpoint and seek to understand how and when they were memorized.

**2.19% of the sequences are memorized.** Of the 1,205,572 sequences we tested for the OLMo model, 44,559 were memorized by at least one checkpoint, but only 26,423 (2.19%) were memorized at the final checkpoint. These examples were

---

[2]This resulted in a marginal decrease of 0.2%, implying that the duplication rate in the overall dataset is quite low.

[3]The memorization rate is a function of many variables, including the length of the prompt used to extract a response (Carlini et al., 2023) and thus we should not extrapolate raw memorizations rates of LLMs without specifying the corresponding prompt lengths.

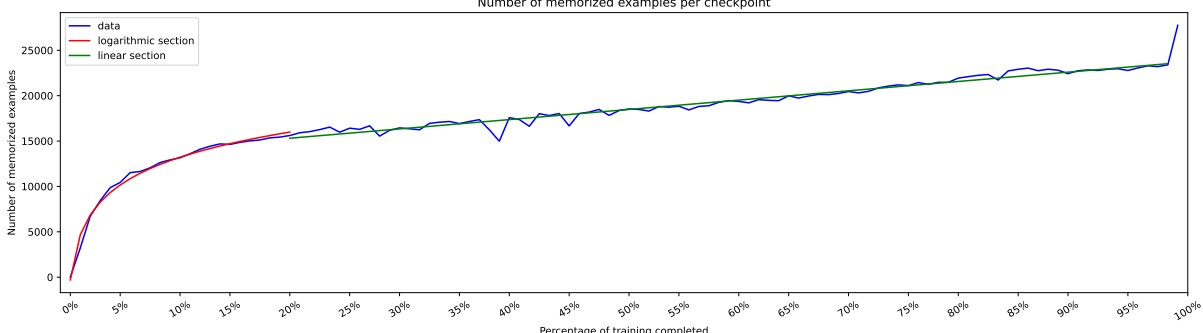

Figure 2: Number of examples memorized by the final LM that were also memorized at each prior checkpoint: logarithmic growth, then linear, followed by a spike.

not all memorized for the first time by the final checkpoint itself; most were memorized earlier and these examples were accumulated over the course of the training process. To understand this phenomenon, we start by plotting how many examples memorized at the final checkpoint were memorized at each prior checkpoint, as seen in Figure 2.

**Growth is logarithmic, then linear, then spikes.** Figure 2 contains three distinct sections, which is representative of the memorization dynamics of the final model. The first 20% of the data display what appears to be logarithmic growth in the number of memorized examples at each checkpoint. Then, for the last 80%, there appears to be fixed, linear growth in the number of memorized examples, with some noise. At the last checkpoint, there is a large spike in the number of memorized examples.

Since at each checkpoint, the model is exposed to a fixed amount of data (22b tokens per 5k training steps), a higher proportion of data the model is exposed to gets memorized during the first section and last step than during most of the training. This provides early evidence for one of our conclusions: sensitive data should be put in the second section, where the memorization rate is the lowest.

### 4.3 Memorizing and Forgetting

We can further explore the memorization dynamics by plotting the "*memorization delta*" at each checkpoint, i.e. the difference between the number of examples memorized at each checkpoint compared to the previous one. Results are shown in Figure 3.

Figure 3 paints a clear picture: the memorization rate decays, then stabilizes at a slightly positive value, and finally spikes at the last checkpoint. In the first 20%, each checkpoint has an average of 665.86 more examples memorized than the last

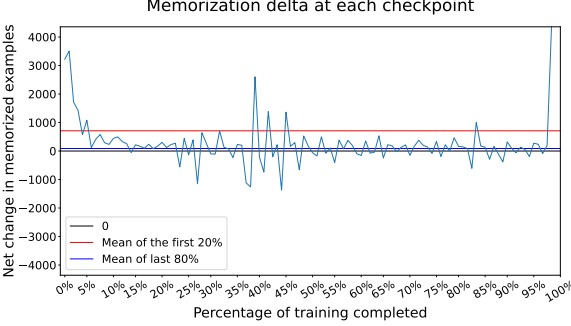

Figure 3: Memorization delta at each checkpoint, defined as the difference in the number of examples memorized compared to the previous checkpoint.

checkpoint. Then for the last 80% (excluding the last step), the memorization delta is only 86.63 examples on average. But at the final checkpoint, 4,177 more examples are memorized than at the previous checkpoint.[4]

**Memorization rate is nearly constant, but forgetting is not.** We go a step further and decompose the memorization delta at each stage into two components: the number of "newly memorized" examples and the number of forgotten examples at each checkpoint compared to the prior checkpoint. We calculate the number of "newly memorized" examples by taking the examples memorized at each checkpoint and checking whether they were memorized at the previous checkpoint as well. Similarly, we calculate the number of forgotten examples by taking the memorized examples at the prior check-

---

[4]This increased growth does indicate anything special about the last checkpoint. The OLMo authors do not specify that step 555,000 in the training was any different than the previous steps. And indeed, our results in Section 4.5 show that if you filter to only examples memorized at any given checkpoint, it appears that that checkpoint has memorized a disproportionate number of examples. This phenomenon is discussed more in that section.

point and seeing how many of them are not memorized at the current checkpoint. Subtracting the number of examples forgotten by each checkpoint from the number of examples newly memorized by each checkpoint is equivalent to the memorization delta in Figure 3. The result of plotting the newly memorized and forgotten examples at each checkpoint is shown in Figure 1.

Figure 1 illustrates what causes the decay in memorization rate early on in the training process: **it is not that these checkpoints have newly memorized more examples, rather, they have forgotten fewer examples.**[5]

It is also interesting to notice what appear to be symmetries in the new memorizations and forgetting rates: for many of these checkpoints when memorization goes up, forgetting goes down, and vice versa. This is the cause of drops and rebounds (prominent around 40% of the way through training) visible in Figure 2, as well as the drops and spikes visible in Figure 3. More investigation is needed to understand the mechanisms that cause these drops and spikes.

The fact that these trends are symmetrical rather than correlated implies that some checkpoints see a relatively higher rate of forgetting paired with a relatively lower rate of memorization (and vice versa) than their neighbors. The fact that rebounds in total memorization follow drops in leads to tentatively conclude that temporarily lowered memorization and raised forgetting make room for rapid consolidation of new memorizations.

### 4.4 Re-memorization

The definition of "forgetting" used in Figure 1 does not imply that no future checkpoint will re-memorize the example. Indeed, because for this plot we filtered to only include examples that are memorized at the final model, every example that is "forgotten" at a previous checkpoint has definitionally been re-memorized later on, or else it would not be present in this dataset. This implies the phenomenon we mentioned previously: **examples are generally memorized early, sometimes forgotten, and often re-memorized later on**.

We investigate this phenomenon by plotting when each example that is memorized by the fi-

---

[5]If the high memorization rate early on was caused by lots of new examples being memorized, we would see the blue line starting high and then decaying to meet the red line. Instead, we see the red line starting low and then growing logarithmically to meet the blue line.

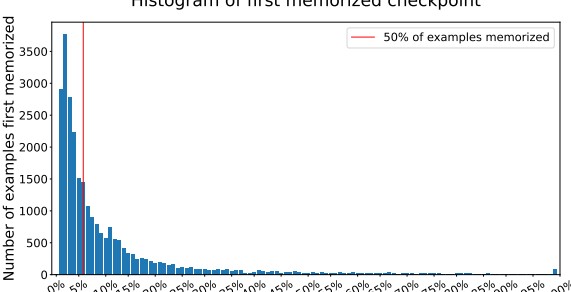

Figure 4: The number of examples that are memorized for the first time at each checkpoint.

nal checkpoint was *first* memorized, as shown in Figure 4. There is a significant skew in the distribution of checkpoints at which examples are first memorized. Of the 26,423 examples memorized by the final checkpoint, 50% of them were memorized within the first 6% of training. While we can see a small spike in the number of examples first memorized by the last checkpoint, the majority of examples that are memorized by the final model are actually first memorized very early in the training process. This is different than the behavior observed in (Stephenson et al., 2021), which found that, in computer vision models, memorization occurs more frequently in later in the training.

**Model re-memorizes many previously forgotten examples.** Figure 4 shows that a majority of examples are first memorized early in training but we know that many of these examples will be forgotten throughout the training process and then re-memorized later. To understand the relation between these phenomena, we also create a plot that shows the start of memorization "streaks" which terminate at the final checkpoint. We define a memorization "streak" for an example as a set of contiguous checkpoints, all of which have memorized that example. To find the beginnings of streaks that end at the final checkpoint, we take all of the examples memorized by the final checkpoint and then work backward, seeing at which checkpoint each example was first memorized within that streak. We then plot the distribution of these streak-start checkpoints, as shown in Figure 5.

Figure 5 is almost a mirror image of the prior plot: while there are 1,127 examples that are memorized continuously throughout the entire training run, the vast majority of examples learned early are forgotten and then re-memorized later on. More than 50% of final streaks are started after 90% of

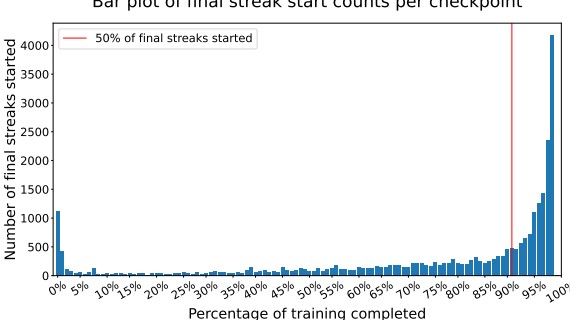

Figure 5: The number of "final streaks" that are started at each checkpoint. See definition in the text.

training is complete, and the final checkpoint alone accounts for more than 15% of the 26,423 examples memorized.

Combining the insights from these two visualizations, we characterize the memorization behavior of models as such: models memorize a great deal of the training data they are initially exposed to, then forget much of it, then re-memorize some of it. It's worth noting that, though it might appear that models re-memorize most of examples close to the end of training, this is actually a statistical artifact: since we are only showing examples memorized by the final checkpoint, there is a bias towards "final streaks" starting near the final checkpoint. This motivates our work in Section 4.5.

**Forgetting and re-memorization happen very frequently throughout training.** While Figure 5 refers only to *final* streaks, there are streaks that end before the final checkpoint. Sometimes, an example will have multiple such streaks, where the first streak represents the first time an example was memorized and each subsequent streak represents a time that example was re-memorized after having been forgotten. We plot the distribution of the number of streaks per example in Figure 6.

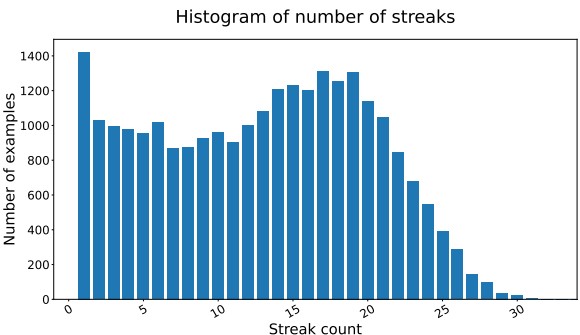

Figure 6: The distribution of streak count per example.

Per Figure 6, while the plurality of examples are memorized only once (left-most bar of the histogram), the bulk of examples are memorized between 15 and 20 times. Since we only looked at 112 checkpoints, this implies that there is a huge amount of forgetting and re-memorization occurring throughout the training process.

**Sometimes examples are "forgotten" because of small changes, while other times they are totally wiped away. In both cases, re-memorization can occur.** We were curious to understand the nature of this forgetting and re-memorization. Are the examples being truly forgotten or is it that the change of a single token resulted in these examples being treated as forgotten, even though most of the semantic information remains intact? Quantitative analysis (discussed in Appendix B) provided no meaningful insight about the nature of re-memorization, so we also analyzed the forgotten and re-memorized examples qualitatively.

Our qualitative analysis showed that the model re-memorized examples that had been only barely forgotten, but that it also re-memorized examples that had been totally forgotten. For instance, the completion "*. With this, we have created a trusted client base, as they are able to easily market their products and services to their best possible customers. Since helps to*" was memorized many times throughout training. It was first memorized in a streak of length one and then immediately forgotten and replaced by the markedly different " *in the market.Technology Data Services, we help you to reach the best target audience who will help your business to grow. We are the leading provider*". For most of the rest of training, the example oscillates between being fully memorized and other markedly different generations. Finally, in the last 32 steps of training, it appears to be "crystallized" (discussed more in Section 4.6), staying continuously memorized, apart from a brief interruption where it minimally changes to "*. With this, we have created a trusted client base, as they are able to easily market their products and services **across the globe without spending much.\nBy***" (emphasis ours) for a single checkpoint before being re-memorized.

The example described in the last section illustrates the phenomena that we observed throughout our qualitative analysis: examples may be markedly forgotten or just barely forgotten, but, in either case, they may get re-memorized. The phenomenon that markedly forgotten examples are

re-memorized is particularly interesting given the low rate of repetition (implied by the extensive deduplication efforts and low rate of duplication in our analysis) because it is not obvious to these authors what could cause a model to re-memorize an example other than being exposed to that example again during training. Further research is needed to understand what causes the phenomenon of re-memorization described here.

## 4.5  Intermediate Checkpoint Analysis

Although the previous analysis focused on the "final" checkpoint, it is important to note that the choice of when to end training is somewhat arbitrary. Although heuristics like the Chinchilla scaling laws (Hoffmann et al., 2022) provide guidance for the compute-optimal amount to train a model, researchers often decide when to stop training based on compute or training data constraints. As such, intermediate checkpoints can be equally useful to analyze. In fact, they provide an opportunity to study an interesting counterfactual scenario: *what would have happened to the examples memorized by the "final" checkpoint if researchers had continued to train the model?*

**Memorization patterns remain the same.**  We arbitrarily select the checkpoint by which 75% training has been completed and filter to only select examples that are memorized at our intermediate model. Reproducing Figure 2, we plot how many of these examples are memorized at each checkpoint in Figure 7.

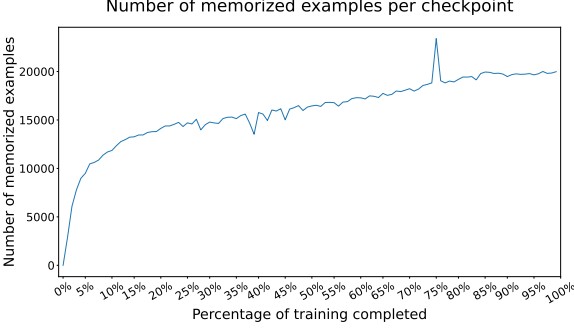
Figure 7: The number of examples memorized by an intermediate checkpoint (75% of the training) that are memorized at each checkpoint.

This plot follows a similar structure to Figure 2, with the same curving growth turning into linear growth. The difference is that, rather than having a spike at the last checkpoint, there is instead a spike and immediate drop which correspond to the checkpoint we are analyzing. This indicates that many of the examples memorized at our checkpoint are memorized *by* that checkpoint (4,722 or 19.12% of the total number of examples memorized at checkpoint 75%) but that many examples are also forgotten at the next stage (4,538 or 18.38%).

**Newly memorized examples are equally likely to stay memorized or get forgotten.**   This raises an interesting question: are the examples forgotten at step 75%+1 primarily examples that the model has just learned at step 75%, or are they examples that the model learned earlier in training? We decompose previous and future states in table 1.

| Step 75% | Newly memorized | Memorized at -1 | Total |
|---|---|---|---|
| **Remain memorized at +1** | 2,409 *9.79%* | 17,740 *71.86%* | 20,149 *81.62%* |
| **Forgotten at +1** | 2,313 *9.38%* | 2,225 *9.01%* | 4,538 *18.38%* |
| **Total** | 4,722 *19.13%* | 19,965 *80.88%* | 24,687 |

Table 1: Previous and future states of examples memorized at 75%. +1/-1 are the next/previous checkpoints.

The vast majority of examples (71.86%) memorized at checkpoint 75% were also memorized at step 75%-1 and remained memorized at step 75%+1. Of examples that were newly memorized at 75%, about half remained memorized at 75%+1 (51.02%) and half were forgotten at step 75%+1 (48.98%). Similarly, of examples that were forgotten at 75%+1, about half were newly memorized at step 75% (50.97%) and about half had also been memorized at step 75%-1 (49.03%).

**Few examples had never been memorized before and few would remain memorized forever.**   Another underlying trend we can analyze by looking at the intermediate checkpoint is the novelty of memorization and the permanence of forgetting. Of the 4,722 examples that were newly memorized at step 75%, only 129 (2.73%) had never been memorized before. Of the 4,538 examples memorized at 75% that are forgotten at 75%+1, only 102 (2.25%) were never memorized again. This all reinforces a key insight of this work: most examples are memorized early, then periodically forgotten and re-memorized throughout the training process.

## 4.6  Crystallization in Early Learning

To further understand how the examples memorized early on are forgotten and re-memorized

throughout training, we examine the examples memorized by an early checkpoint to see how they fare. The very first checkpoint has no memorized examples because it has not been exposed to any training data, so we select the checkpoint after that to better understand the memorization dynamics early in training and see which of those examples remain memorized throughout training (Figure 8).

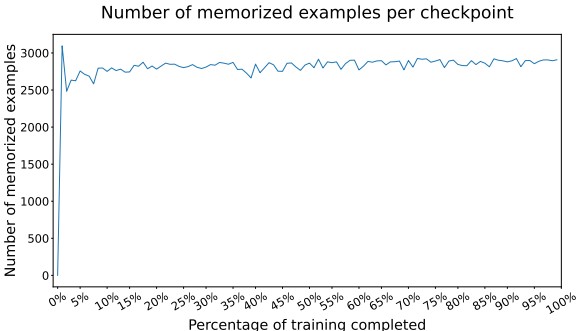

Figure 8: Number of examples memorized by the initial checkpoint that are memorized at each checkpoint.

**Early examples are crystallized.** By plotting the number of examples that were memorized at our initial checkpoint which are also memorized at other checkpoints, we can see a very strong and simple trend: in the first steps of training, 3,096 examples are memorized, and over the course of training, few are forgotten. Notably, very few of these memorized examples are forgotten at the final checkpoint: only 188 (6.07%) of the examples memorized at the initial checkpoint. This implies that examples memorized early on crystalize in the LM's parameters and are unlikely to be forgotten.

We also illustrate this diminishing crystallization by taking the examples memorized at each of the first 10 checkpoints and calculating what proportion of them are continuously memorized for the last 80% of training. We take this ratio to be indicative of the percentage of memorized examples at each checkpoint that are "crystallized" and remain memorized throughout much of training, and plot the results in Figure 9.

Of the 3,096 examples memorized by the first checkpoint, 1,503 (48.55%) are memorized continuously for the last 80% of training. For each subsequent checkpoint, this percentage decays logarithmically, until it reaches a stable forgetting rate at around 20% of examples memorized.

All of this analysis illustrates that, while many examples are forgotten and re-memorized throughout training, the examples memorized early on are

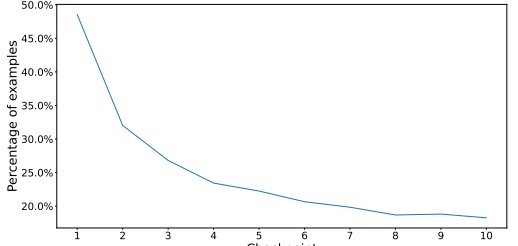

Figure 9: The percentage of examples memorized at each checkpoint that are memorized continuously in the last 80% of training (we call it *crystallization*).

most likely to stay crystallized throughout all of training, while examples memorized later are less likely to be crystallized. This points to the powerful impact of training order on memorization rate.

## 5 Conclusion

Memorization in LLMs is a well-documented phenomenon but more work needs to be done to understand how that memorization occurs, what data is most likely to be memorized, and what can be done to minimize undesirable memorization. This field of research is important for making LLMs useful in commercial applications, as memorization can result in the model leaking private information.

We have made novel contributions by exploring previously unresearched dynamics of memorization throughout the training process. By analyzing memorization at various checkpoints along the training of an LLM, we are able to come to some important conclusions. Most significantly:

1. LMs memorize more earlier on in training

2. LMs forget examples during the training

3. Many forgotten examples are re-memorized

From these conclusions, we tentatively recommend model developers put data which they consider to have a higher likelihood of being sensitive in the middle stages of the training process. In the middle stages, data is memorized at the lowest rate and memorized examples may be forgotten before the model is done being trained.

However, these recommendations can only be tentative because the true test of this hypothesis would be to do controlled experimentation with sensitive data placed at various points in the training process. We hope our work motivates future researchers to perform these experiments to further understand how LLMs memorize.

# Limitations

## 5.1 No Proof of Causality

Ultimately, although our results indicate that there may be an effect of training order on memorization, our experiments are insufficient to prove causality. Because of this, our tentative recommendations can only be fully confirmed by running randomized experiments. For example, although we infer that model developers should put sensitive training data in the middle stages of the training process, it is possible that there are confounding effects that would actually cause this data to be memorized at the same rate, regardless of where it was put. We lack the resources to experiment with training orders but think that our results are sufficient to motivate future investigation into this area.

## 5.2 High Sensitivity

As discussed in Section 2, the method of extracting memorized sequences used in this research is not representative of realistic membership inference attacks. By both prompting the models with exact samples from their training data and using greedy decoding, we maximize the probability that a memorized example will be output. In the real world, attackers are unlikely to have access to the training data and therefore are unlikely to be able to feed it verbatim to the LLM. Additionally, if they did have access to the training data, there would be no purpose in attempting to extract training data from the model. Another factor that contributes to the unrealisticness of this method of attack is that most commercially available LLM providers do not use greedy decoding since it produces highly-repetitive text (Shao et al., 2017).

Although this attack method is unrealistic, we think this area of research is still useful because it allows us to understand all information that is potentially memorized by the model. Since the two things that make this method unrealistic (prompting with exact training data and greedy decoding) also make the model more likely to produce any data it may have memorized, we view our approach as highly sensitive, extracting a large portion of all memorized data, and therefore acting as a canary in the coal mine.

## 5.3 Only English-Language Analysis

As OLMo is a model primarily trained on English text data (Soldaini et al., 2024) and intended for use in English (Groeneveld et al., 2024), very few of the memorized examples we encountered in manual analyses were in languages other than English. There are documented attack vectors that take advantage of low-resource languages to bypass LLM safeguards (Upadhayay and Behzadan, 2024) and it is possible that there are ways to extract training data from LLMs by using low-resource languages. It is also possible that different languages have different memorization dynamics, so further research needs to be performed to understand whether the phenomena we describe are limited to English or would apply to other languages as well.

# Acknowledgements

Our work was funded by the European Research Council (ERC) under the European Union's Horizon 2020 research and innovation programme (grant agreement No. 101019291). This paper reflects the authors' view only, and the ERC is not responsible for any use that may be made of the information it contains.

We would like to thank the ERC for providing this funding. Additional thanks are owed to Marco Baroni for allowing these researchers to use the resources of the Computational Linguistics and Linguistic Theory research group at the Universitat Pompeu Fabra. And special thanks to Vicenç Gómez, the coordinator of the Msc. program in Intelligent Interactive Systems, and Gemma Boleda Torrent, whose passion for linguistics and dedication to her students shines through.

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

## A Pythia

OLMo models are trained using the frequently-used policy of learning rate warmup and annealing, in which the learning rate of an LLM is changed throughout the training process. Specifically, the learning rate is warmed up over the first 5,000 steps and then decayed linearly from there to a tenth of the peak of the learning rate throughout the rest of training. Since (Tirumala et al., 2024) showed that learning rate impacts memorization, we were curious to what extent our results could be explained by changes in the learning rate throughout the training process. As a result, we reproduced our work in OLMo with a similarly-sized Pythia (Biderman et al., 2023) model, which has no learning rate warmup or annealing. We found no significant differences to the trends we described with OLMo.

There are some notable differences between the original OLMo work and the Pythia reproduction, namely:

1. How we sampled the Pythia training dataset (described in the Methodology subsection)

2. The amount of duplication

3. The percentage of memorizations (described in the Results subsection)

4. The classification of different memorized examples (described in the Results subsection)

The fact that, despite all of these differences, the results remained largely the same is heartening evidence that our results generalize to other models.

### A.1 Framework

Pythia models are trained on The Pile (Gao et al., 2020) and, like OLMo, release not only final model weights and the training dataset, but also training methodology and checkpoints. Interestingly, the Pythia models also release their training data in the format and order that it is fed to the model during training, which is not information we were able to find on the OLMo model. As a result, we use a different sampling strategy (as described in Subsection A.2) to extract samples for evaluation. We select the 6.9 billion parameter version of Pythia, since it is the most similar in size to the OLMo model we used.

### A.2 Methodology

Since the authors of Pythia provide the shuffled version of the dataset that they used to train the model, we sampled 1,041,873 examples from evenly-spaced, randomly selected points within the training run, thus ensuring that we would select representative training data. Specifically, we divided Pythia's pre-shuffled Pile dataset's 131,170 iterations into 100 approximately even segments and then selected 10,500 random 64 token sequences from within each segment. We then removed any examples from the same iteration that overlapped as a result of having starts within 64 tokens from each other, resulting in our total of 1,041,873 examples. We then split the 64-token sequences in half, as with OLMo.

The Pythia model has 144 checkpoints separated by 1,000 training steps, starting from step0 and terminating at step143000. We used all of these checkpoints. There are also log-spaced checkpoints provided between step0 and step1000 but we chose not to incorporate these since we wanted evenly-spaced checkpoints.

### A.3 Results

On the whole, we find our results with Pythia to be nearly identical to our results with OLMo, taking into account some differences caused by sampling noise. Notably, the trends we see in OLMo tend to be less pronounced but still present for Pythia.

We have included Pythia reproductions of all of the major figures we used for our OLMo analysis without comment.

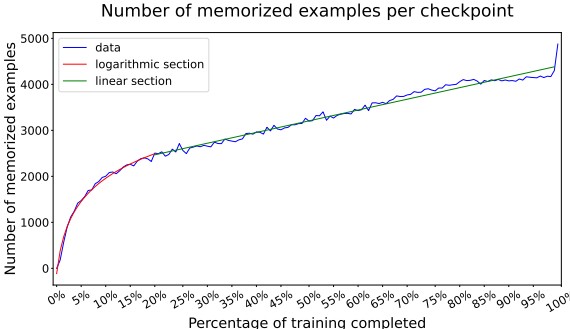

Figure 10: The number of examples memorized by the final checkpoint that are also memorized at each previous checkpoint.

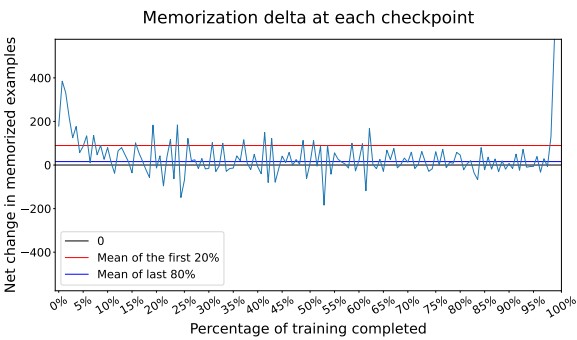

Figure 11: The memorization delta at each checkpoint.

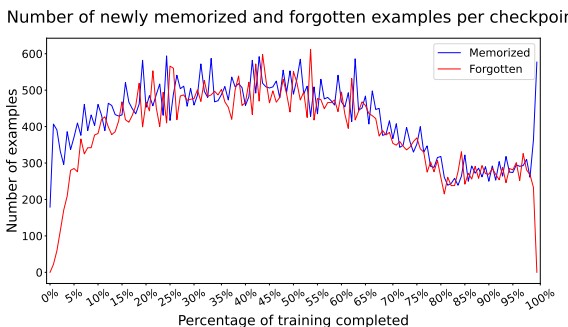

Figure 12: The number of newly memorized and forgotten examples at each checkpoint.

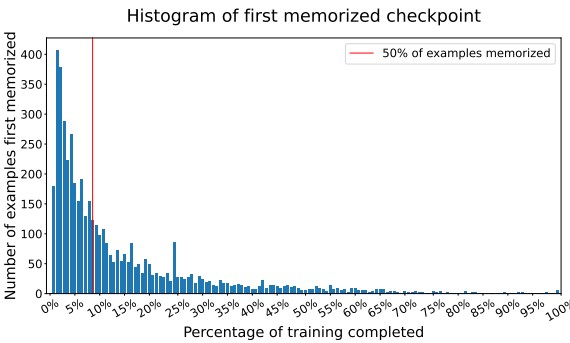

Figure 13: The number of examples that are memorized for the first time at each checkpoint.

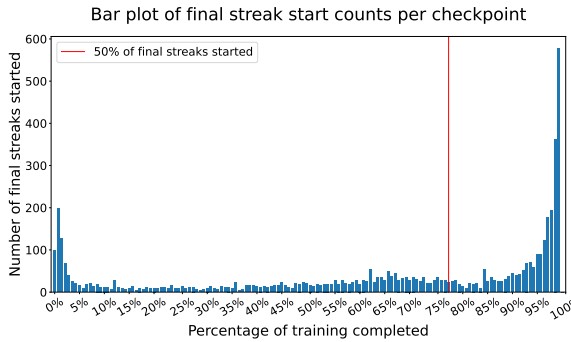

Figure 14: The number of "final streaks" that are started at each checkpoint.

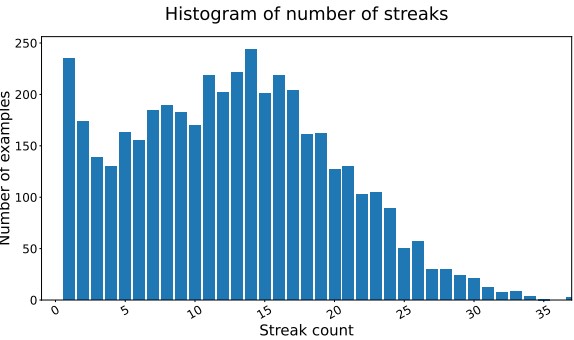

Figure 15: The distribution of streak count per example.

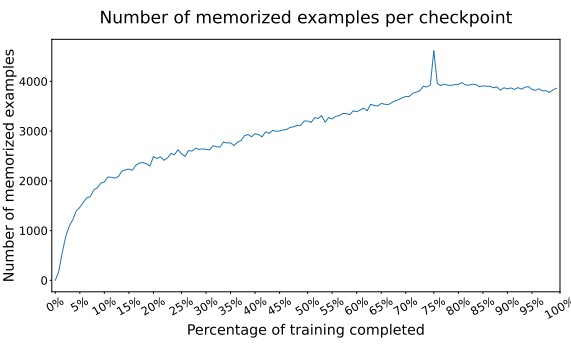

Figure 16: The number of examples memorized by an intermediate checkpoint that are memorized at each checkpoint.

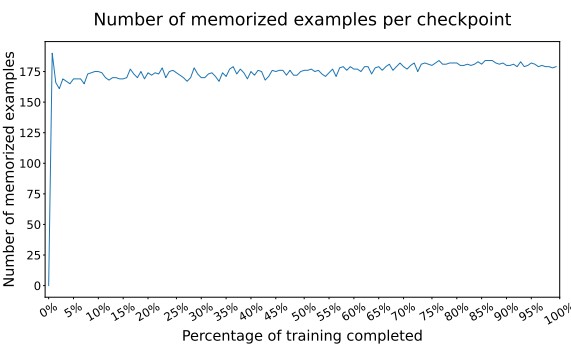

Figure 17: The number of examples memorized by the initial checkpoint that are memorized at each checkpoint.

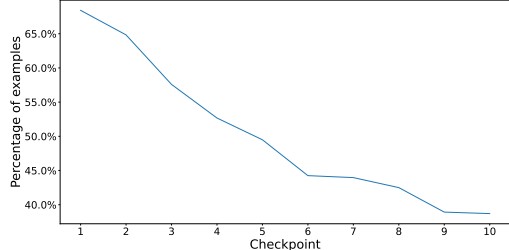

Figure 18: The percentage of examples memorized at each checkpoint that are memorized continuously in the last 80% of training.

## B  Soft Memorization Metrics

All prior work on memorization that we surveyed used a strict definition of extractability, i.e. checking whether the generated output exactly matched the continuation of the sequence in the training data. This is a convenient metric to use because it is reliably easy to evaluate without human intervention. However, for the goal of evaluating undesirable semantic memorization, this is an overly strict definition of "memorization". Therefore, rather than only evaluate extractability according to the "hard" definition (previously defined in the Section 2), we also propose a "soft" definition of extractability: A string $s$ is **$d$-extractable with $k$ tokens of context** from a model $f$ if there exists a (length-$k$) string $p$, such that the concatenation $[p \parallel s]$ is contained in the training data for $f$, and $f$ produces a string which is at most distance $d$ away from $s$ when prompted with $p$ using greedy decoding, for some specified distance measure.

To evaluate whether relaxing the definition of memorization by using d-extractability changed our observed memorization dynamics, we calculated Hamming distance and Levenshtein distance as well as the longest common subsequence for each of the generations in the training dataset.

1. **Hamming distance**: the number of characters that need to be changed in place to make two equal-length sequences identical (Hamming, 1950)

2. **Levenshtein distance**: the number of characters that need to be inserted, deleted, or substituted to make two sequences identical (Levenshtein, 1965)

3. **Longest common subsequence similarity**: the length of the longest continuous sequence of characters that two sequences have in common (Maier, 1978)

We used the Python package "textdistance" (Liferenko, 2024) to efficiently evaluate these similarity metrics.

Performing the same analysis that we had done previously required discretizing these continuous distance metrics, i.e. selecting a value for d. We decided to select these values based on the distribution of each distance metric and arbitrarily selected 2.5%, 5%, 10%, 15%, and 25% quantiles for this cutoff. For example, the 5% quantile represents a cutoff which will treat 5% of the examples as memorized. We reproduced Figure 2 for all three distance measures and all five quantiles in Figure 19.

For all three similarity measures we evaluated, and for all five quantiles, the shapes of the graphs were not meaningfully different than the shape we saw when using the hard definition of memorization: a logarithmic increase followed by a linear increase. When we experimented with different cutoffs, we found that the same shape was generally preserved, except for cutoffs that represented extreme relaxations of the memorization criterion.

We wanted to further investigate whether a different cutoff could help us better understand the memorization dynamics. To do this, we calculated the cutoff values for all 0.01% increments of the cutoff quantiles, and plotted the cutoff values against the percentage of examples that would be treated as "memorized" if we used that cutoff. The results are in Figure 20.

The lack of meaningful inflection points in the graph indicates that these metrics are best understood as continuous measures, rather than being discretized. The first inflection point happens at 2.19%, at which point the cutoff is greater than 0. At a cutoff of 0, the soft memorization metric is equivalent to the hard memorization metric, because the generated text has 0 distance from the expected text. Therefore, this inflection happens at 2.19% because that is the memorization rate according to the hard definition. The other inflection point happens at the 99% mark, which we do not find relevant to this analysis because we do not consider a memorization rate of 99% to be meaningful. Lacking meaningful inflection points indicates to us that there is no trivial and meaningful definition of memorization.

A limitation of all of the metrics we examined is that they do not capture the semantic content of the generations, only making character-wise comparisons. In future work, we hope to further explore meaningful relationships between the definition of memorization used and the trends observed in memorization and forgetting phenomena.

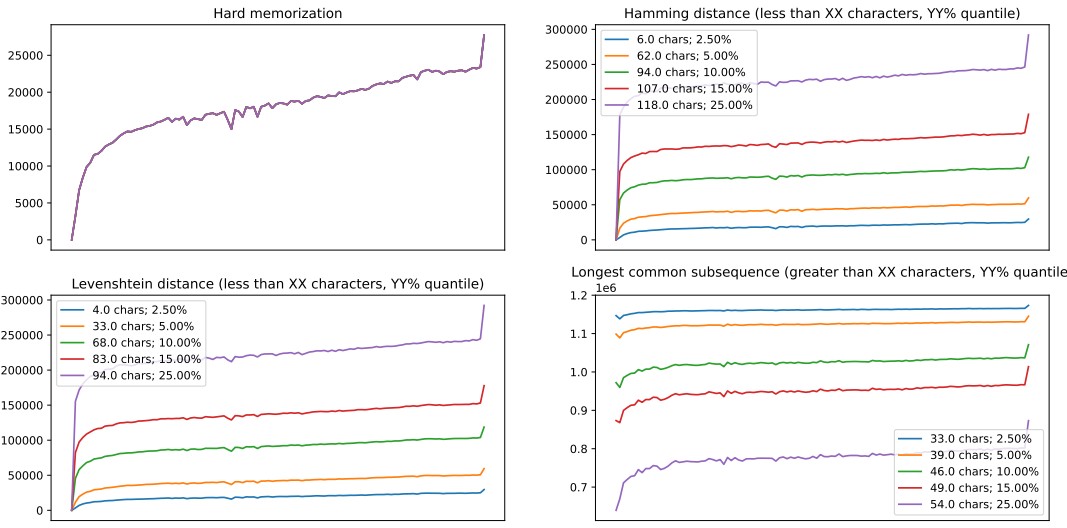

Figure 19: The number of examples that qualify as "memorized" at each checkpoint, using a variety of distance measures and cutoffs.

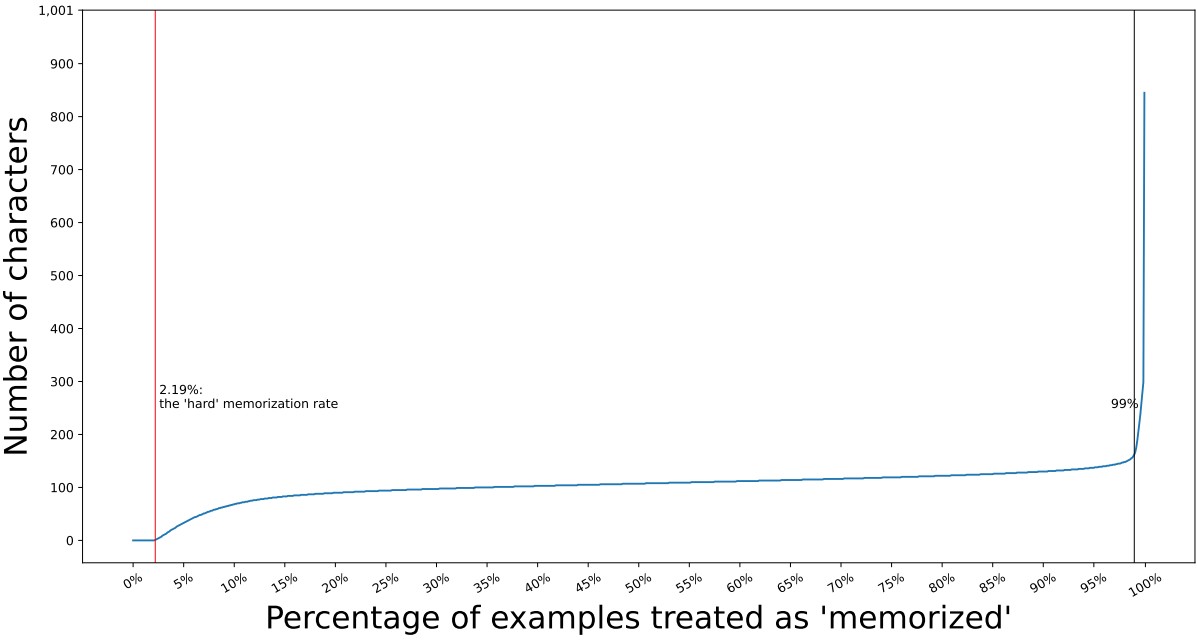

Figure 20: For a percentage of examples to be treated as "memorized", this plots the corresponding number of characters that would be used for a cutoff for Levenshtein distance.