# OpenReview forum: "Learning, Forgetting, Remembering: Insights From Tracking LLM Memorization During Training"
_EMNLP/2024/Workshop/BlackBoxNLP — BlackboxNLP 2024_

### Official Review · Reviewer_ChLs · 2024-09-03

**Overall Assessment:** 5
**Confidence:** 5

**Best Paper:**

3

**Best Paper Justification:**

This is an interesting, well-written, well-executed, and insightful paper. It is one of the best analysis papers I've read recently.

**Comments Questions Suggestions And Typos:**

This is concurrent work, but worth knowing about: https://arxiv.org/abs/2406.11813

- L.31 missing space before citation
- L.222 extra, after the parenthesis
- L.222, what 1127 examples were memorized at each checkpoint? Can you include some of them in the appendix? It may be interesting to analyze them further
- L.257 what causes the spike at the end of training?

**Paper Summary:**

It's one of the best analysis papers I’ve read in the past year.

This paper studies memorization and forgetting during training. It focuses on one of the standard memorization definitions: a model extracting a suffix given a prefix from the training data. Using the different checkpoints, it then measures a model's ability to generate these suffixes during training.
The authors present multiple very interesting insights about such phenomena; for instance, the beginning and end of training are memorized the most, while the middle of training doesn’t tend to memorize as much, providing empirical suggestions on where to put sensitive data, if needed.

**Summary Of Strengths:**

- Well executed paper that studies an interesting problem of memorization and forgetting across training
- Well written
- Providing concrete and empirical suggestions regarding when to include more sensitive data that shouldn’t be memorized

**Summary Of Weaknesses:**

None of these are crucial but rather nitpicky.

- Some terms/presentations can be improved. For instance, some of the figures show the number of examples on the y-axis, but I think it would be easier to digest if you used percentages instead.
- The paper doesn’t discuss the data too much besides duplicates. They mention this a little bit, but perhaps the authors can be more about duplicates that are beyond their exact measurements (which, I believe, include the 64 tokens?)

I understand, though, that this is only the first paper to examine these questions, and they can be explored in future work.

---

### Official Review · Reviewer_QhPi · 2024-09-10

**Overall Assessment:** 2
**Confidence:** 4

**Best Paper:**

1

**Best Paper Justification:**

N/A

**Comments Questions Suggestions And Typos:**

Why not make all plots using the set of all examples that are memorized by any checkpoint? That would avoid biasing results towards any particular checkpoint

Have you examined the "crystallized" examples learned early in training? Are they low-complexity examples? Do they actually occur in the early training batches?

The qualitative analysis of whether re-memorized examples are "barely forgotten" is good, but some quantitative analysis of this question would be even better. Perhaps you could look at something like token-level accuracy during teacher-forcing (k-extractability corresponds to 100% accuracy under teacher forcing).

You can use \citep{paper1,paper2} to cite multiple papers inside one group of parentheses (e.g., lines 097-098).

**Paper Summary:**

This paper analyzes language model memorization behavior across many pre-training checkpoints, focusing on the OLMo and pythia models. They find that examples are frequently memorized and then forgotten over the course of training. They also provide evidence that the rate of memorization is higher at the beginning and end of training, although I have concerns about the validity of these conclusions.

My biggest concern is that one of the main findings seems to be an artifact of the way the authors generate their plots. Most of the results look at the subset of examples that are memorized by the final checkpoint. This creates the illusion that models have higher memorization rates at the very end of training (cf. sentence 2 of the abstract), but this effect goes away when looking at examples memorized by an intermediate checkpoint (Figure 7). I think the claims of the paper are written in a misleading way.

The paper also says that the memorization rate is higher early in training, which leads it to suggest putting sensitive data in the middle of the training run rather than at the beginning. I think there is a fallacy here--there is no evidence of a *causal* relationship between seeing data early and memorizing it. It is possible that the data memorized early is very low-complexity data such as data that has many repeated characters/words (e.g., the "reconstructed" data as described in https://arxiv.org/pdf/2406.17746). Moreover, the paper does not analyze when each example was seen during training, only when it was memorized, so I don't think we can make any claims related to data order.

**Summary Of Strengths:**

* Broad analysis of memorization at different checkpoints
* Shows that forgetting and re-memorization happen frequently

**Summary Of Weaknesses:**

* Large memorization spike at final checkpoint seems to be an artifact of the analysis, but is mentioned prominently in the abstract
* Unclear what the early memorized examples are, no causal link established with data order (despite claims in the paper about data order)

---

### Official Review · Reviewer_gsgQ · 2024-09-11

**Overall Assessment:** 4
**Confidence:** 4

**Best Paper:**

1

**Best Paper Justification:**

N/A

**Comments Questions Suggestions And Typos:**

- It would be interesting to see how sequences forgotten by the final model (but memorized by some earlier checkpoint) are distributed across the previous checkpoints, just as the authors do with examples that are memorized by the final model.
- Do the authors think that the ordering of data will be an important factor even if training was done across multiple epochs?

**Paper Summary:**

This paper investigates how models regurgitate data seen during training verbatim. The authors find that sequences can undergo cycles of memorization and forgetting. They also find the final model is more likely to retain sequences memorized earlier on and toward the end of training than sequences exposed to the model during the middle of training. Based on this insight, the authors recommend placing sensitive data in the middle stages of the training process. While the paper conducts an interesting exploration of memorization training dynamics and presents some interesting findings, the fact that models do not regurgitate exact training sequences at any point in training cannot be taken as evidence for "forgetting" previously memorized content.

**Summary Of Strengths:**

- The paper addresses a timely and interesting research question, exploring the nuances of how language models memorize information over training
- The experiments are well-designed and paper is well-written and easy to follow.
- The topic of memorization in language models has been garnering a lot of attention these days. This paper presents insightful findings on how models memorize information during training

**Summary Of Weaknesses:**

- Based on the findings, it is recommended that sensitive data be placed in the middle during training. I have concerns about the practicality of this claim since it involves knowing ahead of time what might be sensitive information in the training data. And if it is known, in what situations would you still need to include this in training (i.e. why not just exclude the sensitive data altogether)?
- To be rigorous about the core finding in this paper, one would have to try multiple different data orderings and always come to the conclusion that data seen during the earlier/later iterations are more likely to be memorized. However, I understand that this can be a computationally expensive experiment but I would like to see it acknowledged in the limitations. An alternative would be to reproduce similar results in other models (when available).

---

### Decision · Program_Chairs · 2024-09-19

**Decision:**

Accept

**Comment:**

All reviewers appreciated the work on memorization during training and the paper's writing and presentation. One reviewer had a major concern regarding the key findings of memorization rising at the end of training, in light of this effect going away when analyzing an earlier checkpoint. The authors should carefully consider this issue in the camera ready version, and adjust their claims accordingly.
There were also some limitations about drawing causal conclusions, which would be good for the authors to discuss.